# Mortality and Alcohol as Its Cause—Comparative Characteristics of the Two Neighboring Countries: Ukraine and Poland

**DOI:** 10.3390/ijerph182010810

**Published:** 2021-10-14

**Authors:** Oleh Lyubinets, Marta Kachmarska, Katarzyna Maria Sygit, Elżbieta Cipora, Jaroslaw Grshybowskyj

**Affiliations:** 1Department of Public Health, Danylo Halytsky Lviv National Medical University, 79010 Lviv, Ukraine; g.pulmo@gmail.com (O.L.); grshybowskyj@yahoo.de (J.G.); 2Department of Psychiatry and Psychotherapy, Danylo Halytsky Lviv National Medical University, 79010 Lviv, Ukraine; kachmarskamarta@gmail.com; 3Faculty of Health Science, Calisia University, 62-800 Kalisz, Poland; 4Medical Institute, Jan Grodek State University in Sanok, 38-500 Sanok, Poland; elacipora@interia.pl

**Keywords:** mortality, alcohol, working population, Poland, Ukraine

## Abstract

This paper presents a comparative assessment of mortality in Poland and Ukraine, including due to alcohol consumption, by sex, place of residence, and age groups. Mortality from alcohol consumption is and remains one of the health problems of the state’s population. The aim of this study was to establish the difference in mortality, including due to alcohol consumption, in the two neighboring countries. The analysis was conducted in 2008 and 2018 according to statistical institutions in Poland and Ukraine. Data from the codes of the International Statistical Classification of Diseases of the 10th edition: F10, G31.2, G62.1, I42.6, K70, K86.0, and X45 were used to calculate mortality due to alcohol consumption. The share of mortality caused by alcohol consumption in Ukraine in 2008 was 3.52%, and 1.83% in 2018. At the same time, in Poland, there is an increase in this cause of death from 1.72% to 2.36%. Mortality caused by alcohol consumption is the main share of mortality in the section “Mental and behavioral disorders” in both Ukraine, at 73–74%, and Poland, at 82–92%. Changes in the mortality rate in the cities and villages of Ukraine and Poland showed different trends: Poland nated, a significant increase in mortality, while in Ukraine it has halved on average. Overall and alcohol mortality rates in both countries were higher among the male population. The analysis of mortality among people of working age showed that the highest proportion of deaths from alcohol consumption in both countries was among people aged 25–44. Despite the geographical proximity, and similarity of natural and climatic characteristics and population, mortality rates in each country reflect the difference in the medical and demographic situation, and the effectiveness of state social approaches to public health.

## 1. Introduction

The medical and demographic situation in Ukraine is negative, and is characterized by a decrease in the birth rate for the period 2014–2019 by 19.4% and an increase in the mortality rate by 6.16 per 100 thousand population [1,2]. Against the background of negative natural growth, there has also been a decrease in the average age of women giving birth and increased morbidity and perinatal mortality [3].

Stably high mortality rates in Ukraine compared to those in European countries require constant monitoring and identification of the causes of this phenomenon. In Ukraine, as in other countries, noncommunicable diseases remain the leading cause of morbidity, disability, and premature mortality. The most effective way to reduce them is to prevent the development of non-communicable diseases by addressing the behavioral risk factors underlying them at the population and individual levels: smoking, alcohol consumption, excessive salt intake, low physical activity, overweight and obesity, and unhealthy diet [4].

Alcohol is one of the most important health risk factors and causes of premature mortality. According to Jürgen Rehm and co-authors, alcohol consumption caused 14.6% of all premature adult mortality in eight European countries, 17.3% in men and 8.0% in women [5]. The study of medical and social areas of alcohol consumption is valuable. Thus, 83.54 ± 0.98% of the 1422 students surveyed in Ukraine consume alcoholic beverages, three-quarters of them choose mostly alcoholic beverages with low alcohol content. The main motive for drinking low-alcohol beverages is hedonism. The majority of respondents who consume low-alcohol beverages do not consider the occurrence of alcohol dependence when consuming alcohol (56.85 ± 1.44%), which indicates an insufficient level of preventive measures [6].

The issue of the impact of morbidity and mortality on alcohol consumption on the economic sphere of the state is important. Thus, Błażej Łyszczarz notes that the total production losses associated with alcohol deaths in the EU in 2016 amounted to 32.1 billion euros [7]. Expenditure per capita (share of gross domestic product expenditure) was EUR 62.88 (0.215%) for the EU as a whole and ranged from EUR 17.29 (0.062%) in Malta to EUR 192.93 (0.875%) in Lithuania.

Macroeconomic fluctuations can affect alcohol problems in different ways in socio-demographic groups. The Spanish crisis after 2008 was a turning point for such conclusions: the annual changes in the percentage of deaths directly from alcohol mortality were 6.9% in 2002–2007 and 3.7% in 2008–2011 among the employed, and −4.3% and −0.4%, respectively, among the unemployed [8].

Despite the WHO’s emphasis on the importance of restricting alcohol consumption through fiscal measures, the governments of Lithuania (in 1999), Poland (in 2002), and Finland (in 2004) reduced the excise tax on alcoholic beverages by, respectively 44%, 30%, and 44%. These decisions led to immediate and impressive health consequences. In Poland, the decline in vodka prices wasaccompaniedby a sharp increase in registered alcohol sales from 7 L per capita in 2002 to almost 10 L per capita in 2008 and an annual increase in alcohol-related mortality, especially for menaged 45–64 years [9].

Given the geographical proximity of Poland and Ukraine—sharing a common border of 535 km—similarity of natural and climatic characteristics, similarity of population (as of 1 January 2019, at 38.4 and 42.3 million people, respectively), different political and economic situation, and insignificant number of publications in international scientific journals comparing mortality rates in neighboring countries, including Ukraine, and the impact of alcohol on the health of the population, the present study examined the mortality rates of the population of Ukraine and Poland and its level due to alcohol consumption.

## 2. Objective

This study aims to identify trends and features of changes in the overall mortality rate, as well as mortality due to alcohol consumption, in two countries with different economic development, Ukraine and Poland, for a ten-year period—in 2008 and 2018.

## 3. Materials and Method

Regarding the analytical strategy of the study, we assessed whether there are any basic differences between the two geographically neighboring countries—Poland and Ukraine, in terms of overall mortality and alcohol-related mortality.

Data on population mortality were taken from the database of the State Statistics Service of Ukraine and the Central Statistical Office of the Republic of Poland.

Statistical analysis of the obtained data was performed by measuring the ratio and interval measurement (2008–2018) with zero data for 2008 in each of the countries, and in comparison between countries, in the respective year.

To calculate the mortality rate due to alcohol use, data from the codes of the International Statistical Classification of Diseases of the 10th revision were used:

Chapter V Mental and behavioral disorders (F00-F99):

F10Mental and behavioral disorders due to use of alcohol.

Chapter VI Diseases of the nervous system (G00-G99):

G31.2Degeneration of nervous system due to alcohol;

G62.1Alcoholic polyneuropathy.

Chapter IX Diseases of the circulatory system (I00-I99):

I42.6Alcoholic cardiomyopathy.

Chapter XI Diseases of the digestive system (K00-K93):

K70Alcoholic liver disease;

K86.0Alcohol-induced chronic pancreatitis.

Chapter XX External causes of morbidity and mortality (V01-Y98):

X45Accidental poisoning by and exposure to alcohol.

To be able to compare the obtained data on the mortality rate of the working-age population, we determined its range of 16–65 years, as according to the Organization for Economic Cooperation and Development, in 2016, the average normal retirement age was 64.3 years for men and 63.7 years for women in all schemes and countries [10]. In Ukraine, this age is 60 years.

## 4. Results

The basis for comparing mortality rates in 2008 and 2018 in Poland and Ukraine was the geographical proximity of these countries, a common border, the similarity of natural and climatic characteristics, and similarity of population. At the same time, the political and economic situation of these countries differs sharply. Poland has been a part of the European Union since 2004, which played a significant role in the development of the state and contributed to obtaining significant investments for the economy and social sphere, including health systems. Its GDP per capita in 2008 was USD 13,996; in 2018 this was already equal to USD 15,468.4 (according to the World Bank). The growth over these 10 years was 10.5%.

Ukraine, as an independent state, has been taking steps since 1991 to deepen cooperation with European countries in the economic and social spheres. The Government and the President of Ukraine have set the task of gaining membership in the European Union. This became especially relevant after the beginning of Russia’s aggression in 2014. Ukraine’s GDP from 2008 to 2018 decreased by 20.3%, from 3887.2 to 3096.8 USD. In comparison with the data for 2018, Poland’s GDP was 5 times higher than in Ukraine.

Thus, the overall mortality rate in Ukraine for 10 years decreased by 14.76% (according to the State Statistics Service of Ukraine) [11]. In Poland, the opposite changes took place—there was an increase in overall mortality by 8.33% (according to Statistics Poland) [12]. At the same time, the overall mortality rate in Ukraine was, in 2008 and 2018, higher than that of Poland.

The overall mortality rate of Ukraine in 2015 was one of the highest (after Moldova, Kazakhstan, Kyrgyzstan, and Georgia) among the countries that are members of the Word Health Organization (WHO) Regional Office for Europe [13]. Comparisons of mortality rates in Ukraine and Poland per 1000 people by main chapters of causes of death are presented in Table 1.

Alcohol is one of the most important determinants of health and the cause of premature mortality. We analyzed the mortality rates in Ukraine and Poland according to the codes specified in the section “Materials and methods of research”. Comparison of the share of mortality caused by alcohol consumption, in the structure of total mortality in certain chapters of causes of death by chapters and codes of the International Statistical Classification of Diseases and changes in mortality in 2008 and 2018, are shown in Table 2.

Place of residence and analysis of causes of death by sex are important criteria in analyzing the level of mortality caused by alcohol consumption in a particular chapter of causes of death (Table 3 and Table 4).

The analysis of the causes of death caused by alcohol consumption by sex and among residents of cities and villages of Ukraine and Poland had a clear difference.

Given the importance of the human factor as the main labor force of the country, we calculated the total mortality rate of the working-age population (Table 5), as well as analyzed the mortality rates for three age groups, such as 16–24 years, 25–44 years, and 45–64 years (Table 6, Table 7 and Table 8).

## 5. Discussion

Comparison of medical and demographic indicators in different countries or regions of the world is one of the elements of an analytical approach to solving public health problems. In addition to WHO data, i.e.,Global Health Observatory Data Repository and Framework for alcohol policy in the WHO European Region, this kind of analysis is presented in many works, including M. Marmot et al., F. Baum, J. P. Mackenbach, C. Bambra et al., and further Polish researchers [14,15,16,17,18,19,20]. Comparison of data from some geographically close regions of different countries was carried out in the study of J.Grshybowskyj and co-authors, who found a significant difference in the development processes in terms of medical and demographic indicators [21].

Our analysis of the data showed that the main causes of death in Ukraine in 2008 and in 2018 were (in % of all causes of death) (Figure 1):Diseases of the circulatory system—63.64 and 66.98;Neoplasms—11.80 and 13.43;External causes of morbidity and mortality—8.14 and 5.28;Diseases of the digestive system—4.67 and 4.18;Diseases of the respiratory system—3.09 and 2.22.

It is established that, for these 10 years in Ukraine, there was an increase in the share of deaths from diseases of the circulatory system (by 3.34%) and tumors (by 1.63%).

In Poland over the years, the main causes of death were the same pathology, but in contrast in Ukraine, mortality from tumors and respiratory diseases was higher (in %% of all causes of death):Diseases of the circulatory system—45.84 and 40.55;Neoplasms—25.18 and 26.38;External causes of morbidity and mortality—6.68 and 4.85;Diseases of the respiratory system—5.09 and 6.65;Diseases of the digestive system—4.50 and 4.18.

It should also be noted that, in ten years, Poland saw a decrease in the mortality rate from diseases of the circulatory system and external causes, while the share of deaths due to respiratory diseases (+1.57%), tumors (+1.21%), endocrine diseases, eating disorders, and metabolic disorders (+0.48%), mental and behavioral disorders (+0.36%) and diseases of the nervous system (+0.27%) all increased.

When comparing mortality rates by main classes of causes in Ukraine and Poland (Table 1), it was found that the total mortality in Ukraine exceeded the figures in Poland, both in 2008 and in 2018, by 1.64 and 1.29 times, respectively (16.31 and 13.90 in Ukraine per 1000 people and 9.95 and 10.78 in Poland), including excess mortality observed in some infectious and parasitic diseases (5.28 and 4.20 times, respectively), diseases of the circulatory system (2.29 and 2.12 times), and external causes (1.99 times in 2008). In contrast, the mortality rate in Poland was higher in endocrine diseases, eating disorders and metabolic disorders (3 and 5 times, respectively) and neoplasms (1.31 and 1.53 times, respectively). From 2008 to 2018, there was a significant increase in the difference in mortality from respiratory diseases—with almost the same figures in 2008 to 2.32 times more in Poland in 2018.It is necessary to note separately the changes that have occurred in the mortality rates of mental and behavioral disorders—in 2008, the mortality rate in Ukraine was slightly higher than in Poland (0.062 in Ukraine against 0.054 in Poland), and in 2018 the death rate in Poland from mental and behavioral disorders exceeded that in Ukraine 2.9 times (0.034 in Ukraine and 0.098 in Poland). Similar, but less pronounced changes were observed in mortality from diseases of the nervous system.

Mortality from pathology caused by alcohol consumption occupies an ambiguous share in the structure of population mortality [22,23]. In the period we are comparing, the share of mortality caused by alcohol consumption in Ukraine decreased from 3.52 to 1.83%. At the same time, in Poland, there was an increase in this cause of death from 1.72 to 2.36%. When comparing the share of such causes of death in our selected chapters, it was found that growth was observed in mental and behavioral disorders due to alcohol use (in Ukraine), andalcoholic liver disease, accidental poisoning by and exposure to alcohol, and alcohol-induced chronic pancreatitis (in Poland). Moreover, the increase in mortality from alcoholic liver disease exceeded 13% and, in Ukraine, there was a decrease in such indicatorsfrom 14.52% to 8.71%.

It should be noted that the mortality caused by alcohol consumption was the main share of mortality in the class of mental and behavioral disorders: 73% in Ukraine and 92% in Poland 2008, and 74% and 83%, respectively, in 2018 (Table 2).

Place of residence is an important criterion in analyzing the level of mortality caused by alcohol consumption in a particular chapter of causes of death, as it allows for appropriate organizational and preventive measures for medical care in the area, whether in the city or in the countryside. Studies of changes in mortality rates in cities and villages of Ukraine and Poland have shown different trends in mental and behavioral disorders (F10) and alcoholic liver disease (K70).

In 2008, mortality from mental and behavioral disorders due to alcohol consumption among peasants was higher in Ukraine than in Poland (0.071 vs. 0.044) (Table 3). However, in 2018 it became much higherin Poland: among urban residents 5.7 times (0.014 in Ukraine, 0.080 in Poland) and among rural residentsmore than 2 times (0.039 in Ukraine, 0.083 in Poland). For this reason, mortality in both urban and rural areas in Ukraine has halved on average, while in Poland the mortality rate has increased by the same amount. The mortality rate due to alcohol and behavioral disorders due to alcohol consumption in Ukraine is predominant among rural residents; in Poland it is slightly more common among urban residents.

Mortality due to alcoholic liver disease among urban residents of Poland from 2008 to 2018 increased from 0.093 to 0.157 per 1000 inhabitants, and in 2018 was 4 times higher than that among residents of Ukrainian cities (0.104 and 0.039, respectively). The same dynamics are observed among rural residents.

One of the reasons for this difference in the mortality rates of rural and urban populations of Ukraine is the problem of unemployment, especially in rural areas, and it acquires new features [24]. Despite the fact that many unemployed rural residents have their own farm as a source of livelihood, this way of employment, as well as work in agricultural production in general, which is mostly low-paid and uncreative, can provide neither material wealth nor the need for self-realization in accordance with the demands of modern young villagers. Therefore, living in the countryside today is characterized not only by physical overload but often by life’s frustrations and the associated alcoholism, which shortens life. Therefore, the mortality of the rural population ofworking age is higher than in the urban [25,26]. Similar problems are observed in other countries [27,28].

Analysis of the causes of death by sex showed that the overall mortality rate and mortality caused by alcohol consumption in both Ukraine and Poland was higher among the male population in 2008 and 2018 (Table 4). The comparison of indicators makes it possible to state that, against the background of declining mortality in Ukraine and its growth in Poland, the most pronounced change was the increase in mortality caused by alcohol consumption among women in Poland, which exceeded that among women in Ukraine (in 2018,0.097 in Ukraine, 0.101 in Poland).

J. Mackenbach, N. Ryngach, and V. Kruchanytsia [29,30,31] indicate a high death rate from alcohol-related causes, especially in men. The effects of alcohol consumption are largely associated with smoking [32,33,34,35,36].

Among the male population of Ukraine by certain chapters of causes of death due to alcohol consumption, the highest mortality rate in 2008 and 2018 was observed for alcoholic cardiomyopathy (0.334 and 0.163 per 1000 cases, respectively) and for accidental poisoning and alcohol exposure (0.329 and 0.134 per 1000 cases, respectively). In Poland, the highest mortality rates for men were observed from alcoholic liver disease (0.125 and 0.213 per 1000 cases, respectively, in 2008 and 2018) and from mental and behavioral disorders (0.091 and 0.146, respectively).

However, the mortality of the male population of Ukraine caused by alcohol consumption, in the chapter “Mental and behavioral disorders” was actually the most common, and there was a tendency to increase the share of mortality in the chapter: 82.79% in 2008 and 83.57% in 2018. In Poland, the same negative dynamics of mortality weredue to alcoholic liver disease (23.12% in 2008 and 39.05% in 2018), and the highest proportion of deaths caused by alcohol consumption was in the chapter “Mental and behavioral disorders” (96.22% in 2008 and 92.29% in 2018). For other chapters of causes of death, the share in the mortality of the population caused by alcohol consumption was much lower.

Among the female population with a similar ratio of mortality to mortality in the corresponding chapter in Poland, the highest mortality rate with a tendency to increase is determined in subchapter K.86.0 “Alcoholic liver disease”: 0.035 per 1000 cases in 2008 and 0.072 in 2018. When comparing these indicators with the data in Ukraine, they were twice lower in 2008 and already exceeded them in 2018 (0.065 per 1000 cases in 2008 and 0.030 in 2018).

As among the male population, among women in Ukraine and Poland in the chapter “Mental and behavioral disorders”, the highest percentage of deaths was caused by alcohol consumption. However, it was twice lower than among men (Ukraine: 45.39% in 2008 and 46.89% in 2018; Poland: 69.28% in 2008 and 49.40% in 2018).

Given the importance of the human factor as the main labor resource of the country, we calculated the indicators of total mortality and mortality caused by alcohol consumption in working age (Table 5). It is established that with the overall decrease in the mortality of the working population from 2008 to 2018 in both Ukraine and Poland, mortality due to alcohol consumption in Ukraine decreased by 55.35%, and in Poland, it increased by 35.70%. Accordingly, the share of deaths due to alcohol consumption in the total number of deaths changed: in Ukraine, their share decreased from 9.77% to 5.82%, and in Poland it increased from 5.11% to 7.66%.

The same trends were observed for the mortality chapter of the working population due to mental and behavioral disorders: a significant decrease in both total and alcohol-related mortality in Ukraine (by 52.2% and 51.3%, respectively) and its increase in Poland (by 45.1% and 43.9%). In general, mortality rates in this chapter of causes in Poland have become higher than in Ukraine.

Another chapter where the same mortality trends are observed was chapter XI: Diseases of the digestive system. With the prevalence of the overall mortality rate in Ukraine over the mortality rate in Poland for this chapter of causes of death (2.9 and 1.9 times in 2008 and 2108, respectively), there were significant changes in the mortality rate due to alcoholic liver disease: a 54% decrease in mortality in Ukraine, and growth in Poland by 66.4%. In general, the share of deaths due to alcohol use in this chapter of causes of death in Ukraine decreased from 17.86% to 11.58%, and in Poland increased from 31.20% to 52.58%, exceeding the mortality rate in Ukraine in 2018.

According to the other classes of causes of death, it is necessary to state a significant predominance of the mortality rate in Ukraine against that in Poland due to alcoholic cardiomyopathy and due to degeneration of the nervous system caused by alcohol consumption.

The analysis of mortality rates by three age groups, 16–24 years, 25–44 years, and 45–64 years, showed that mortality was predominant among people aged 45–64 (Table 6). With the general decrease in the mortality rate of the working-age population both in Poland and in Ukraine, it was most pronounced among persons aged 16–24 (19.9% in Poland and 42.6% in Ukraine). Regarding mortality due to alcohol consumption, its level in all age groups was higher in Ukraine than in Poland, except in 2018 in the age group 45–64 years, when the figures were almost the same. The highest share of deaths due to alcohol consumption in those of working-age was among persons aged 25–44, with tendenciesin Poland to increase (from 8.10% to 12.22%) and in Ukraine to decrease (from 12.89% to 9.28%).

When comparing the data on mortality rates by age groups, and by the defined chapters of causes of death, it was found that the main cause of death in the age groups 16–24 and 25–44 years in both countries were external causes of morbidity and mortality. However, in these groups in Ukraine, the mortality due to alcohol consumption was mainly due to “Accidental poisoning and exposure to alcohol” (0.022 per 1000 cases in 2008 and 0.006 in 2018). In Poland, this was observed only in 2008 among the working population aged 16–24 (0.003 per 1000 cases). In 2018, in this age group, and in 2008 and 2018 among residents aged 25–44, the highest mortality due to alcohol consumption was due to alcoholic liver disease (2008: 25–44 years—0.050 per 1000 cases; 2018: 16–24 years—0.003 and 25–44 years—0.088).

In both countries, in the age group 45–64 years, the main cause of death was “Diseases of the circulatory system”in 2008 and 2018.

The age group 45–64 years, as a non-mobile productive group of the population, accounted for a significant share of the population of Poland in 2018—43.8%, which indicates an aging population [37]. In Ukraine, it was equal to 27.4%. In this group of the population in Ukraine, the main cause of death in 2008 and 2018 was “Diseases of the circulatory system”; in Poland this was “Neoplasms”.

In Poland, the main causes of death from alcohol use in this age group were alcoholic liver disease and mental and behavioral disorders (respectively: 2008: 0.199 and 0.127; 2018: 0.310 and 0.180) (Table 7). When comparing the indicators for 2008 and 2018, there is a 55.69%increase in mortality due to alcoholic liver disease and a41.07% increase in mortality due to mental and behavioral disorders due to alcohol consumption.

In all age groups, the proportion of deaths due to alcohol use in the total number of deaths by chapters of causes of death was highest in the chapter “Mental and behavioral disorders”. This was higher than 95% among people aged 25–44 and 45–64.

In Ukraine, alcohol consumption has become the most common cause of death due to alcoholic cardiomyopathy (Table 8). In 2018, the mortality rate from alcoholic cardiomyopathy decreased by more than 2 times, but continued to be the leading cause of death due to alcohol consumption, and exceeded this figure among the working population of Poland by 29.5 times. As in Poland, the share of deaths due to alcohol use in the total number of deaths by chapters of causes of death was the highest in the chapter “Mental and behavioral disorders” in all age groups, and reached the maximum among persons aged 45–64 in 2008—91.3%, and in 2018—89.0%.

## 6. Conclusions

Despite the geographical proximity, the similarity of natural and climatic characteristics, and similarity of population, mortality rates in each country reflect the difference in the medical and demographic situation, and the effectiveness of state social approaches to public health. Alcohol plays and continues to play an important role in the mortality rate: the share of mortality caused by alcohol consumption in Ukraine in 2008 was 3.52%, and 1.83% in 2018. At the same time, in Poland, there was an increase in such a cause of death from 1.72% (2008) to 2.36%. Mortality caused by alcohol consumption is the main share of mortality in the chapter on mental and behavioral disorders. Changes in the mortality rate in cities and villages of Ukraine and Poland showed different trends: in Poland, during the study period there was a significant increase in mortality due to alcohol consumption to a greater extent among the rural population. At the same time, mortality in Ukraine has halved on average among both urban and rural residents. The overall mortality and alcohol-related mortality rates in both Ukraine and Poland were higher among the male population. Analysis of mortality among people of working age showed that the highest proportion of deaths due to alcohol consumption in both countries was among people aged 25–44. The main cause of death in this and the group of 16–24 years was due to accidental poisoning and alcohol exposure. In the age group 45–64 years in Poland, the main causes of death with a tendency to increase were alcoholic liver disease and mental and behavioral disorders due to alcohol consumption. In Ukraine, the most common cause of death was alcoholic cardiomyopathy.

## Figures and Tables

**Figure 1 ijerph-18-10810-f001:**
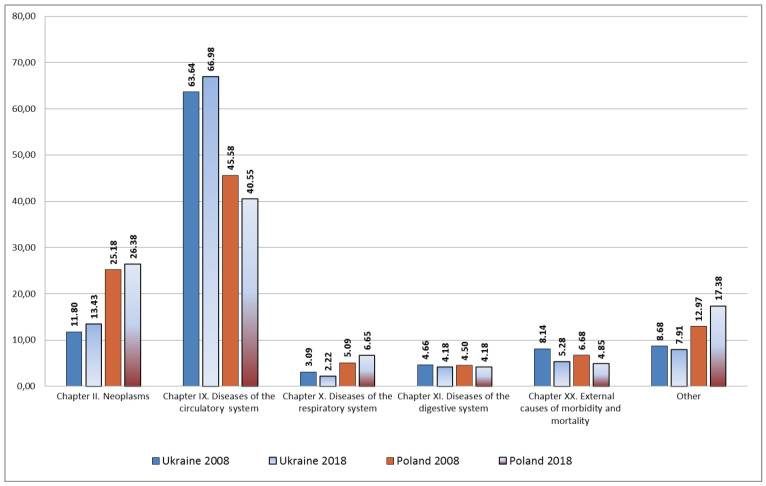
The main causes of mortality in Ukraine and Poland in 2008 and 2018 (in %).

**Table 1 ijerph-18-10810-t001:** Mortality rates in Ukraine and Poland per 1000 people by main classes of causes of death.

The Main Classes of Causes of Mortality	Ukraine			Poland		
	Rate 2008	Rate 2018	% Change	Rate 2008	Rate 2018	% Change
The overall mortality rate	16.31	13.90	−14.76	9.95	10.78	+8.33
Chapter I. Certain infectious and parasitic diseases	0.37	0.21	−43.16	0.07	0.05	−32.39
Chapter II. Neoplasms	1.92	1.86	−3.38	2.51	2.85	+13.53
Chapter IV. Endocrine, nutritional, and metabolic diseases	0.06	0.05	−13.11	0.18	0.25	+36.41
Chapter V: Mental and behavioral disorders	0.06	0.03	−51.61	0.06	0.10	+78.18
Chapter VI. Diseases of the nervous system	0.16	0.11	−29.11	0.13	0.17	+30.77
Chapter IX. Diseases of the circulatory system	10.38	9.28	−10.64	4.54	4.37	−3.64
Chapter X. Diseases of the respiratory system	0.50	0.31	−38.77	0.51	0.72	+41.90
Chapter XI. Diseases of the digestive system	0.76	0.58	−23.92	0.45	0.45	+0.67
Chapter XX. External causes of morbidity and mortality	1.33	0.73	−44.91	0.67	0.52	−21.35

% Change—decrease or increase rate from 2008 to 2018 in percent (of observed values).

**Table 2 ijerph-18-10810-t002:** Change in the proportion of mortality caused by alcohol consumption in certain classes of causes of death.

Mortality Due to Alcohol Consumption		Ukraine		Poland	
		% in a Certain Class of Causes of Death	Change 2008–2018	% in a Certain Class of Causes of Death	Change 2008–2018
All causes	2008	3.52	−1.69	1.72	+0.64
	2018	1.83		2.36	
Chapter V: F10 Mental and behavioral disorders due to use of alcohol	2008	72.99	+1.11	91.91	−9.08
	2018	74.10		82.83	
Chapter VI: G31.2 Degeneration of nervous system due to alcohol	2008	22.78	−8.94	1.07	−0.92
	2018	13.84		0.15	
Chapter VI: G62.1 Alcoholic polyneuropathy	2008	0.33	−0.22	0.08	−0.05
	2018	0.11		0.03	
Chapter IX: I42.6 Alcoholic cardiomyopathy	2008	1.88	−0.88	0.15	−0.08
	2018	1.00		0.07	
Chapter XI: K70 Alcoholic liver disease	2008	14.52	−5.81	17.55	+13.57
	2018	8.71		31.12	
Chapter XI: K86.0 Alcohol-induced chronic pancreatitis	2008	0.20	−0.10	0.20	+0.16
	2018	0.10		0.36	
Chapter XX: X45Accidental poisoning by and exposure to alcohol	2008	14.00	−4.00	5.07	+0.39
	2018	10.00		5.46	

**Table 3 ijerph-18-10810-t003:** Mortality caused by alcohol consumption by certain classes of causes of death depending on the place of residence.

Mortality Rate	Place of Residence	UKRAINE			POLAND			% Change* 2008	% Change* 2018
		Rate 2008	Rate 2018	% Change	Rate 2008	Rate 2018	% Change		
Overall mortality rate	urban	14.75	12.51	−15.21	9.81	11.06	+12.68	−33.50	−11.62
	rural	19.83	17.19	−13.29	10.17	10.36	+1.91	−48.71	−39.72
The mortality rate due to alcohol	urban	0.50	0.20	−59.72	0.19	0.27	+41.16	−62.17	+32.57
	rural	0.73	0.37	−49.08	0.14	0.23	+64.32	−80.66	−37.60
Chapter V. Mental and behavioral disorders	urban	0.05	0.02	−57.33	0.06	0.10	+70.74	+25.21	+401.07
	rural	0.10	0.05	−46.02	0.05	0.10	96.23	−48.50	+87.22
F10 Mental and behavioral disorders due to use of alcohol	urban	0.03	0.01	−56.59	0.05	0.08	+47.07	+62.74	+451.39
	rural	0.07	0.04	−45.39	0.04	0.08	+90.80	−38.40	115.22
Chapter VI. Diseases of the nervous system	urban	0.15	0.10	−34.54	0.14	0.19	+40.02	−7.98	+96.82
	rural	0.18	0.15	−19.20	0.12	0.14	+15.45	−33.71	−5.29
G31.2 Degeneration of nervous system due to alcohol	urban	0.032	0.012	−63.82	0.002	0.0003	−83.18	−95.19	−97.76
	rural	0.045	0.024	−45.70	0.001	0.0003	−77.25	−97.43	−98.92
G62.1 Alcoholic polyneuropathy	urban	0.0004	0.0001	−70.57	0.0001	0.003	−32.71	−63.28	−16.05
	rural	0.0009	0.0002	−82.63	0.0001	0.000	-	−92.37	-
Chapter IX. Diseases of the circulatory system	urban	9.14	8.06	−11.81	4.30	4.32	+0.68	−53.01	−46.35
	rural	13.15	12.10	−7.92	4.91	4.44	−9.64	−62.62	−63.32
I42.6 Alcoholic cardiomyopathy	urban	0.18	0.08	−56.30	0.01	0.003	−55.14	−96.03	−95.93
	rural	0.24	0.13	−45.54	0.01	0.002	−61.74	−97.31	−98.11
Chapter XI. Diseases of the digestive system	urban	0.79	0.57	−27.42	0.50	0.49	−0.59	−36.62	−13.18
	rural	0.72	0.61	−15.34	0.37	0.38	+3.79	−48.51	−36.88
K70 Alcoholic liver disease	urban	0.10	0.04	−62.40	0.09	0.16	+68.74	−10.01	+303.80
	rural	0.13	0.08	−39.33	0.06	0.11	+105.41	−56.02	+48.90
K86.0 Alcohol-induced chronic pancreatitis	urban	0.0014	0.0005	−65.67	0.001	0.002	+61.48	−23.51	+259.80
	rural	0.0018	0.0008	−52.23	0.001	0.001	+112.67	−61.84	+69.88
Chapter XX. External causes of morbidity and mortality	urban	1.20	0.65	−46.77	0.61	0.48	−20.79	−49.96	−25.54
	rural	1.59	0.93	−41.28	0.76	0.59	−22.34	−52.31	−36.93
X45Accidental poisoning by and exposure to alcohol	urban	0.16	0.06	−61.50	0.03	0.03	−19.93	−78.79	−55.88
	rural	0.25	0.10	−59.00	0.03	0.03	−8.49	−86.24	−69.29

% Change—decrease or increase rate from 2008 to 2018 in percent (of observed values). % Change*—decrease or increase rate from Ukraine to Poland in percent (of observed values).

**Table 4 ijerph-18-10810-t004:** Mortality rate of men and women due to alcohol consumption.

Mortality Rate	Sex	Ukraine			Poland			% Change* 2008	% Change* 2018
		Rate 2008	Rate 2018	%Change	Rate 2008	Rate 2018	%Change		
Overall mortality rate	Men	18.19	14.89	−18.14	10.99	11.49	+4.60	−39.60	−22.83
	Women	14.82	13.15	−11.24	8.98	10.11	+12.57	−39.38	−23.12
Overall mortality rate caused by alcohol consumption	Men	0.98	0.44	−54.98	0.29	0.39	+33.91	−69.86	−10.35
	Women	0.24	0.10	−58.82	0.06	0.10	+76.50	−75.76	+3.91
Chapter V. Mental and behavioral disorders	Men	0.10	0.05	−52.24	0.09	0.16	+66.45	−4.55	+232.65
	Women	0.03	0.01	−52.74	0.02	0.04	+148.46	−44.15	+193.61
F10 Mental and behavioral disorders due to use of alcohol	Men	0.08	0.04	−51.79	0.09	0.15	+59.66	+10.93	+267.38
	Women	0.014	0.006	−51.18	0.01	0.02	+77.16	−14.77	+209.29
Chapter VI. Diseases of the nervous system	Men	0.23	0.15	−34.80	0.13	0.16	+21.48	−43.42	+5.41
	Women	0.10	0.08	−18.11	0.13	0.18	+39.59	+34.07	+128.52
G31.2 Degeneration of nervous system due to alcohol	Men	0.06	0.03	−55.78	0.0023	0.0004	−83.87	−96.21	−98.62
	Women	0.014	0.005	−62.22	0.0005	0.0002	−70.18	−96.45	−97.20
G62.1 Alcoholic polyneuropathy	Men	0.0010	0.0002	−84.44	0.0002	0.0001	−50.47	−78.03	−30.04
	Women	0.0001	0.0001	−26.71	0.00	0.00	-	-	-
Chapter IX. Diseases of the circulatory system	Men	10.04	8.79	−12.39	4.43	4.13	−6.74	−55.88	−53.04
	Women	10.75	9.76	−9.18	4.64	4.60	−0.86	−56.85	−52.90
I42.6 Alcoholic cardiomyopathy	Men	0.33	0.16	−51.32	0.013	0.006	−56.95	−96.15	−96.60
	Women	0.08	0.03	−56.93	0.001	0.000	−66.87	−98.63	−98.95
Chapter XI. Diseases of the digestive system	Men	1.07	0.78	−26.65	0.54	0.55	+0.92	−49.31	−30.25
	Women	0.51	0.41	−19.11	0.36	0.36	−0.02	−28.45	−11.56
K70 Alcoholic liver disease	Men	0.16	0.07	−54.91	0.12	0.21	+70.43	−23.98	+187.38
	Women	0.07	0.03	−53.37	0.04	0.07	+103.47	−45.78	+136.59
K86.0 Alcohol-induced chronic pancreatitis	Men	0.003	0.001	−52.48	0.002	0.003	+77.63	−39.17	+127.38
	Women	0.0006	0.0000	−92.67	0.0003	0.0005	+65.64	−49.60	+1039.08
Chapter XX. External causes of morbidity and mortality	Men	2.29	1.27	−44.74	1.05	0.81	−23.13	−54.17	−36.25
	Women	0.51	0.27	−46.32	0.31	0.26	−15.74	−40.33	−6.34
X45Accidental poisoning by and exposure to alcohol	Men	0.33	0.13	−59.32	0.06	0.03	−56.70	−81.50	−80.31
	Women	0.06	0.02	−67.25	0.008	0.007	−11.52	−87.09	−65.14

% Change—decrease or increase rate from 2008 to 2018 in percent (of observed values). % Change*—decrease or increase rate from Ukraine to Poland in percent (of observed values).

**Table 5 ijerph-18-10810-t005:** Mortality working age population.

Mortality Rate	Ukraine			Poland			% Change* 2008	% Change* 2018
	Rate 2008	Rate 2018	% Change	Rate 2008	Rate 2018	% Change		
Overall mortality rate	7.97	5.97	−25.10	4.24	3.84	−9.46	−46.81	−35.70
Overall mortality rate caused by alcohol consumption	0.78	0.35	−55.35	0.22	0.29	+35.70	−72.16	−15.40
% of deaths due to alcohol consumption in the total number of deaths	9.77	5.82	-	5.11	7.66	-	−47.66	+31.57
Chapter V. Mental and behavioral disorders								
Mortality rate	0.07	0.03	−52.19	0.06	0.09	+45.08	−11.05	+169.90
Deaths on mental and behavioral disorders due to use of alcohol Mortality due to mental and behavioral disorders due to alcohol consumption	0.06	0.03	−51.28	0.06	0.09	+43.94	+3.44	+205.62
% of deaths due to alcohol use in this class of causes of death	84.22	85.82	-	97.94	97.17	-	+16.29	+13.23
Chapter VI. Diseases of the nervous system								
Mortality rate	0.18	0.11	−39.31	0.07	0.06	−2.72	−63.39	−41.31
Mortality due to degeneration of the nervous system caused by alcohol consumption	0.048	0.020	−58.32	0.002	0.000	−83.82	−96.50	−98.64
Mortality due to alcohol field neuropathy	0.0007	0.0002	−75.76	0.0002	0.0000	−73.99	−79.35	−77.85
% of deaths due to alcohol use in this class of causes of death	27.37	18.68	-	2.81	0.49	-	−89.74	−97.37
Chapter IX. Diseases of the circulatory system								
Mortality rate	2.86	2.35	−17.71	1.16	0.88	−23.84	−59.57	−62.58
Mortality due to alcoholic cardiomyopathy	0.264	0.126	−52.39	0.008	0.003	−60.26	−96.99	−97.49
% of deaths due to alcohol use in this class of causes of death	9.24	5.35	-	0.69	0.36	-	−92.55	−93.28
Chapter XI. Diseases of the digestive system								
Mortality rate	0.85	0.60	−29.29	0.32	0.31	−1.24	−62.66	−47.85
Mortality due to alcoholic liver disease	0.15	0.07	−54.00	0.10	0.16	+66.35	−34.61	+136.45
Mortality due to chronic alcoholic pancreatitis	0.002	0.001	−63.93	0.001	0.002	+74.49	−45.88	+161.80
% of deaths due to alcohol use in this class of causes of death	17.86	11.58	-	31.20	52.58	-	+74.69	+353.95
Chapter XX. External causes of morbidity and mortality								
Mortality rate	1.59	0.85	−46.78	0.69	0.49	−28.20	−56.88	−41.82
Mortality due to accidental poisoning and alcohol exposure	0.25	0.10	−59.52	0.05	0.04	−21.70	−82.16	−65.49
% of deaths due to alcohol use in this class of causes of death	15.85	12.06	-	6.56	7.15	-	−58.62	−40.68

% Change—decrease or increase rate from 2008 to 2018 in percent (of observed values). % Change*—decrease or increase rate from Ukraine to Poland in percent (of observed values).

**Table 6 ijerph-18-10810-t006:** Mortality of the working age population by age groups.

Age Groups	Mortality Rate	Ukraine			Poland			% Change* 2008	% Change* 2018
		Rate 2008	Rate 2018	% Change	Rate 2008	Rate 2018	% Change		
16–24	Mortality in this age group	1.18	0.68	−42.58	0.66	0.53	−19.85	1.80	1.29
	Mortality of the population in this age group due to alcohol consumption	0.04	0.01	−79.55	0.01	0.01	5.23	8.80	1.50
	Proportion of deaths due to alcohol consumption in this age group	3.73	1.39	-	0.83	1.10	-	4.46	1.27
25–44	Mortality in this age group	4.87	2.89	−40.53	1.49	1.27	−15.14	3.26	2.29
	Mortality of the population in this age group due to alcohol consumption	0.63	0.27	−57.10	0.12	0.16	28.04	5.19	1.74
	Proportion of deaths due to alcohol consumption in this age group	12.89	9.28	-	8.10	12.22	-	1.59	0.76
45–64	Mortality in this age group	15.03	11.18	−25.60	9.04	8.05	−10.85	1.66	1.39
	Mortality of the population in this age group due to alcohol consumption	1.34	0.55	−59.00	0.43	0.56	31.11	3.12	0.98
	Proportion of deaths due to alcohol consumption in this age group	8.91	4.91	-	4.75	6.98	-	1.88	0.70

%Change—decrease or increase rate from 2008 to 2018 in percent (of observed values). %Change*—decrease or increase rate from Ukraine to Poland (of observed values).

**Table 7 ijerph-18-10810-t007:** Mortality of the working age population in Poland by causes of death and by age groups.

Mortality Rate\Age Groups	16–24 Years			25–44 Years			45–64 Years		
	Rate 2008	Rate 2018	%Change	Rate 2008	Rate 2018	%Change	Rate 2008	Rate 2018	%Change
F10–F99. Chapter V. Mental and behavioral disorders	0.002	0.001	−63.67	0.033	0.043	30.36	0.129	0.185	42.99
F 10. Mortality due to mental and behavioral disorders due to alcohol consumption	0.001	0.001	−58.48	0.032	0.041	30.58	0.127	0.180	41.07
Proportion of deaths due to alcohol consumption in this age group (%)	58.33	66.67	-	96.67	96.83	-	98.64	97.32	-
G00–G99. Chapter VI. Diseases of the nervous system	0.023	0.025	8.69	0.034	0.028	−16.10	0.121	0.119	−1.80
G31.2 Mortality due to degeneration of the nervous system caused by alcohol consumption	0.000	0.000	-	0.001	0.000	-	0.003	0.001	−77.72
G62.1 Mortality due to alcohol field neuropathy	0.000	0.000	-	0.000	0.000	-	0.000	0.000	−74.53
Proportion of deaths due to alcohol consumption in this age group (%)	0.00	0.00	-	3.48	0.00	-	2.89	0.67	-
I00-I99. Chapter IX. Diseases of the circulatory system	0.035	0.018	−49.49	0.249	0.166	−33.15	2.708	2.031	−25.00
I42.6 Mortality due to alcoholic cardiomyopathy	0.000	0.000	-	0.002	0.001	−67.77	0.018	0.007	−60.57
Proportion of deaths due to alcohol consumption in this age group (%)	0.00	0.00	-	0.95	0.46	-	0.67	0.35	-
K00-K93. Chapter XI. Diseases of the digestive system	0.009	0.008	−7.25	0.150	0.150	0.56	0.659	0.618	−6.29
K70 Mortality due to alcoholic liver disease	0.002	0.003	117.97	0.050	0.088	73.36	0.199	0.310	55.69
K86.0 Mortality due to chronic alcoholic pancreatitis	0.000	0.000	-	0.001	0.001	−30.16	0.001	0.004	171.66
Proportion of deaths due to alcohol consumption in this age group (%)	17.02	40.00	-	34.73	58.87	-	30.43	50.84	-
V01-Y98. Chapter XX. External causes of morbidity and mortality	0.470	0.336	−28.62	0.549	0.397	−27.69	0.949	0.664	−29.96
X45 Mortality due to accidental poisoning and alcohol exposure	0.003	0.002	−27.34	0.034	0.024	−28.35	0.079	0.060	−23.66
Proportion of deaths due to alcohol consumption in this age group (%)	0.56	0.57	-	6.16	6.11	-	8.34	9.09	-

**Table 8 ijerph-18-10810-t008:** Mortality of the working age population of Ukraine by causes of death and by age groups.

Mortality Rate\Age Groups	16–24 Years			25–44 Years			45–64 Years		
	Rate 2008	Rate 2018	% Change	Rate 2008	Rate 2018	% Change	Rate 2008	Rate 2018	% Change
F10–F99. Chapter V. Mental and behavioral disorders	0.009	0.002	−82.16	0.066	0.027	−59.36	0.111	0.054	−51.97
F 10. Mortality due to mental and behavioral disorders due to alcohol consumption	0.005	0.001	−82.16	0.050	0.022	−56.64	0.102	0.048	−53.18
Proportion of deaths due to alcohol consumption in this age group (%)	50.00	50.00	-	75.68	80.75	-	91.31	89.01	-
G00–G99. Chapter VI. Diseases of the nervous system	0.051	0.034	−33.11	0.163	0.085	−48.02	0.264	0.160	−39.37
G31.2 Mortality due to degeneration of the nervous system caused by alcohol consumption	0.002	0.000	−89.22	0.039	0.015	−62.35	0.083	0.033	−60.45
G62.1 Mortality due to alcohol field neuropathy	0.000	0.000	-	0.000	0.000	−79.54	0.001	0.000	−75.63
Proportion of deaths due to alcohol consumption in this age group (%)	5.15	0.78	-	24.25	17.49	-	31.83	20.63	-
I00-I99. Chapter IX. Diseases of the circulatory system	0.071	0.054	−23.94	0.897	0.652	−27.33	6.519	5.023	−22.94
I42.6 Mortality due to alcoholic cardiomyopathy	0.006	0.002	−61.68	0.191	0.087	−54.31	0.483	0.210	−56.53
Proportion of deaths due to alcohol consumption in this age group (%)	7.86	3.96	-	21.30	13.39	-	7.42	4.18	-
K00-K93. Chapter XI. Diseases of the digestive system	0.052	0.023	−56.63	0.684	0.401	−41.41	1.468	1.024	−30.28
K70 Mortality due to alcoholic liver disease	0.010	0.001	−94.35	0.130	0.053	−59.20	0.248	0.110	−55.63
K86.0 Mortality due to chronic alcoholic pancreatitis	0.000	0.000	-	0.002	0.001	−48.85	0.003	0.001	−79.28
Proportion of deaths due to alcohol consumption in this age group (%)	18.26	2.38	-	19.33	13.51	-	17.09	10.80	-
V01-Y98. Chapter XX. External causes of morbidity and mortality	0.746	0.427	−42.76	1.526	0.784	−48.63	2.122	0.784	−63.07
X45 Mortality due to accidental poisoning and alcohol exposure	0.022	0.006	−73.95	0.214	0.091	−57.69	0.419	0.106	−74.78
Proportion of deaths due to alcohol consumption in this age group (%)	2.90	1.32	-	14.03	11.56	-	19.74	13.48	-

## Data Availability

Not applicable.

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
