# Peer review of "Mortality and Alcohol as Its Cause—Comparative Characteristics of the Two Neighboring Countries: Ukraine and Poland"

_ijerph, 2021, doi:10.3390/ijerph182010810_

Round 1

Reviewer 1 Report

This study has a high interest to the readers and to the others researchers. The topic of alcohol use is still poorly evaluated and all information is important anywhere in the world. I think that some aspects can be improved in the presentation of the research, such as: organizing the tables according to the publication pattern and exposing all the names of the categories and putting the political and economic issue of the countries in the discussion during the period of analysis. We know that more authoritarian states with less freedom for their citizens tend to have a higher level of health problems, so this political scenario of countries needs to be at least described in the study. 

Author Response

The letter is attached.

Reviewer 2 Report

Thank you for the opportunity to review this paper that examines the change in mortality due to alcohol consumption, in the two neighbouring countries, namely Ukraine and Poland. I read the article with interest and found the topic relevant to the research area. However, I think there are several things that need to be addressed before this paper is suitable for publication.

First of all, I will not be able to comment on specifics in the manuscript, mainly because the results are ambiguous and the method-section are omitting the analytical strategy or description of the statistical tests or models, used to investigate the changes over time and the differences between countries.

More general, I would suggest the authors to rewrite the introduction to better support the conclusion and include a clear comparison of the two countries in question e.g. the GDP, population size, healthcare system etc. Further, the authors should increase the readability of the result-section, for example by using similar figures as Figure 1 on page 13 – which is easy to read and understand. Also, the result section is missing indication or description of statistically significant estimates. In the current form, the discussion section mostly list the results presented in table 2-8.

Author Response

The letter is attached.

Reviewer 3 Report

No comments

Author Response

The letter is attached.

Round 2

Reviewer 2 Report

Thank you for addressing some of the issues raised in the first review-report. I still think that the result-section needs to be revised, before the paper is suitable for publication. Some of the estimates seems redundant, but it is difficult to assess because the arguments for and description of the chosen calculations are lacking.

I have some specific concerns and comments:

In the section MATERIALS AND METHOD the authors write: “Data on population mortality were taken from the database of the State Statistics Service of Ukraine and the Central Statistical Office of the Republic of Poland. Regarding the analytical strategy of the study: we assessed whether there are any basic differences between the two geographically neighboring countries - Poland and Ukraine, in terms of overall mortality and alcohol-related mortality.”

However, it is still unclear how the authors assessed the basic differences? i.e. which estimates/calculation are used to compare the rates for e.g. each cause, each county, each time point?

In the section RESULTS it is unclear why and when the authors calculated % and/or %%. For example in Table 2 on page 5, the ‘Mortality due to alcohol consumption’ seems to be calculated different for the two countries:

UKRAINE: % in a certain class of causes of death - Сhange 2008/2018

POLAND: %% in a certain class of causes of death - Сhange 2008/2018

However in the discussion on page 15, the authors state: “It should be noted that the mortality caused by alcohol consumption was the main share of mortality in the class of Mental and behavioral disorders: in Ukraine - 73-74 %%, in Poland - 92-83%% (Table 2).” - Maybe it is a typing error?

Similar inconsistency is found in Table 8 on page 12

In the section DISCUSSION the authors write: “When comparing mortality rates by main chapters of causes in Ukraine and Poland, it was found that if the total mortality rate in Ukraine exceeds the figures in Poland, both in 2008 and in 2018, respectively by 1.64 and 1.29 times, the mortality rate is exceeded, more than 2 times, was observed in some infectious and parasitic diseases (5.28 and 4.20 times, respectively), diseases of the circulatory system (2.29 and 2.12 times), external causes (1.99 times in 2008). In contrast, the mortality rate in Poland was higher in endocrine diseases, eating disorders, and metabolic disorders (3 and 5 times, respectively) and neoplasms (1.31 and 1.53 times). From 2008 to 2018, there was a significant increase in the difference in mortality from respiratory diseases - with almost the same figures in 2008 to 2.32 times more in Poland in 2018.”

The “times” of increase/decrease in rates, referred to in this paragraph are not mentioned in the RESULTS. If the estimates are publishes in another manuscript, please add the reference for the publication. Otherwise, include the estimates in the results of this manuscript. Please do not introduce new results or estimates in the discussion that are not mentioned in the preceding result-section.
